# Large Language Models powered Neural Solvers for Generalized Vehicle Routing Problems

**Cong Dao Tran[1], Quan Nguyen-Tri[1], Huynh Thi Thanh Binh[2], Hoang Thanh-Tung[3]**
[1]FPT Software AI Center, [2]Hanoi University of Science and Technology,
[3]University of Engineering and Technology, Vietnam National University
{daotc2, quannt40}@fpt.com, binhht@soict.hust.edu.vn, htt210@gmail.com

## Abstract

Neural Combinatorial Optimization (NCO) has shown promise in solving combinatorial optimization problems end-to-end with minimal expert-driven algorithm design. However, existing constructive NCO methods for Vehicle Routing Problems (VRPs) often rely on attention-based node selection mechanisms that struggle with large-scale instances. To address this, we propose a directed fine-tuning approach for NCO based on LLM-driven automatic heuristic design. We first introduce an evolution-driven process that extracts implicit structural features from input instances, forming LLM-guided attention bias. This bias is then integrated into the neural model's attention scores, enhancing solution flexibility and scalability. Instead of retraining from scratch, we fine-tune the model on a small, diverse dataset to transfer learned heuristics effectively to larger problem instances. Experimental results show that our approach achieves state-of-the-art performance on TSP and CVRP, significantly improving generalization to both synthetic and real-world datasets (TSPLIB and CVRPLIB) with thousands of nodes [1].

## 1 Introduction

Vehicle Routing Problems (VRPs) are fundamental NP-hard combinatorial optimization challenges in logistics and public transportation Konstantakopoulos et al. (2022). As problem size increases, solving VRPs becomes increasingly complex and time-consuming, posing significant challenges for traditional exact and heuristic methods. While heuristic approaches like LKH Helsgaun (2017b) and HGS Vidal (2022b) have been widely studied, they often rely on domain-specific knowledge, limiting their generalization and efficiency for large-scale instances.

Neural Combinatorial Optimization (NCO) Bello et al. (2017) has emerged as a promising alternative, leveraging deep learning to learn heuristics and generate solutions via Supervised Learning (SL) Joshi et al. (2019); Chen et al. (2023) or Reinforcement Learning (RL) Kool et al. (2019). However, SL-based methods require vast amounts of labeled data, making them impractical for large-scale problems Bello et al. (2017), while RL-based approaches suffer from sparse rewards, memory constraints, and poor generalization, limiting their effectiveness in real-world applications.

To enhance generalization from small-scale to large-scale VRPs, several strategies have been explored. One approach involves training models on small instances and extending them to larger problems Fu et al. (2021a); Drakulic et al. (2024). Another direction employs divide-and-conquer techniques Hou et al. (2023), while recent advances introduce inductive biases Kim et al. (2022) and novel architectures like heavy decoders Luo et al. (2023), diffusion models Sun & Yang (2023), and meta-learning techniques Son et al. (2023). Ensemble methods, such as GLOP and ELG Ye et al. (2024b); Gao et al. (2024), further enhance generalization by integrating locally transferable topological features. However, many of these methods still rely on expert knowledge, limiting their scalability and adaptability.

---

[1]Our code is available at: https://github.com/Fsoft-AIC/NCO-LLM

With the rise of Large Language Models (LLMs), there is growing interest in their ability to generate heuristics and optimize algorithms Li et al. (2023); Yang et al. (2023). While LLMs have been explored for in-context learning and heuristic selection, standalone LLM-based approaches often struggle to generate novel insights beyond existing knowledge Mahowald et al. (2023). To overcome this limitation, recent studies have integrated LLMs with Evolutionary Computation (EC) to automate heuristic generation Yang et al. (2023); Romera-Paredes et al. (2024); Ye et al. (2024a); Liu et al. (2024), as seen in FunSearch Romera-Paredes et al. (2024), EoH Liu et al. (2024), and ReEvo Ye et al. (2024a).

Inspired by these advancements, we propose a directed fine-tuning strategy that leverages LLM-generated heuristics to improve large-scale generalization in NCO models. Specifically, we augment pre-trained networks with inductive attention bias derived automatically from LLMs through an evolutionary process, enabling the model to extract topological features and enhance its cross-size generalization. This approach fine-tunes the model on a small, diverse dataset, improving solution diversity and flexibility. Notably, our method is model-agnostic, seamlessly enhancing pre-trained NCO attention models without architectural modifications. We evaluate our approach on two canonical VRPs, i.e., Traveling Salesman Problem (TSP) and Capacitated Vehicle Routing Problem (CVRP) on both generated and real-world datasets (TSPLIB, CVRPLIB). Our results demonstrate state-of-the-art performance, significantly improving the generalization capabilities of constructive NCO models.

Our contributions can be summarized as follows:

- We propose a new LLM-guided attention bias for model fine-tuning to enhance the generalization capabilities of neural combinatorial optimization models. This attention bias is automatically designed by LLMs, which are used to augment any attention-based NCO models for better performance on large-scale problems.

- We develop an efficient fine-tuning process that leverages training on instances of varying sizes. This method improves the model's flexibility and solution quality by incorporating LLM-generated attention bias, allowing effective generalization to larger and more diverse problem instances without altering their architecture.

- Our proposed method can achieve state-of-the-art performance in solving the TSP and CVRP across various scales and also generalizes well to solve real-world TSPLib/CVRPLib problems.

## 2 BACKGROUND AND RELATED WORK

### 2.1 VEHICLE ROUTING PROBLEMS

VRPs are *NP-hard* combinatorial optimization challenges with widespread practical applications. A VRP instance consists of a node set $V$ with $n$ nodes, where each node $v_i \in V$ must be visited while minimizing the total travel cost. Variants of VRPs introduce problem-specific constraints, influencing solution complexity. This study focuses on two fundamental VRP variants: the Euclidean Traveling Salesman Problem (TSP) and the Capacitated Vehicle Routing Problem (CVRP). In TSP, the objective is to find an optimal tour visiting each node in $V$ exactly once. CVRP extends TSP by introducing a depot node $v_0$, vehicle capacity constraints, and node-specific demands. A feasible CVRP solution consists of multiple capacity-constrained sub-tours, each starting and ending at the depot while ensuring all non-depot nodes are visited exactly once.

### 2.2 NEURAL COMBINATORIAL OPTIMIZATION

**Constructive Neural Models.** These models are commonly employed for solving VRPs using a Markov Decision Process (MDP) and incorporate a Transformer-based encoder-decoder framework as their policy network Vinyals et al. (2015); Kool et al. (2019); Kwon et al. (2020); Luo et al. (2023). The encoder captures node features, while the decoder generates a tour $\pi$ based on the extracted features and the action history of node selections. During each step $t$ of the MDP, the decoder selects an unvisited node, masking invalid nodes (visited and exceeding capacity) to ensure feasibility until the completion of the tour. Given a VRP instance $\mathbf{s}$, the constructive NCO process

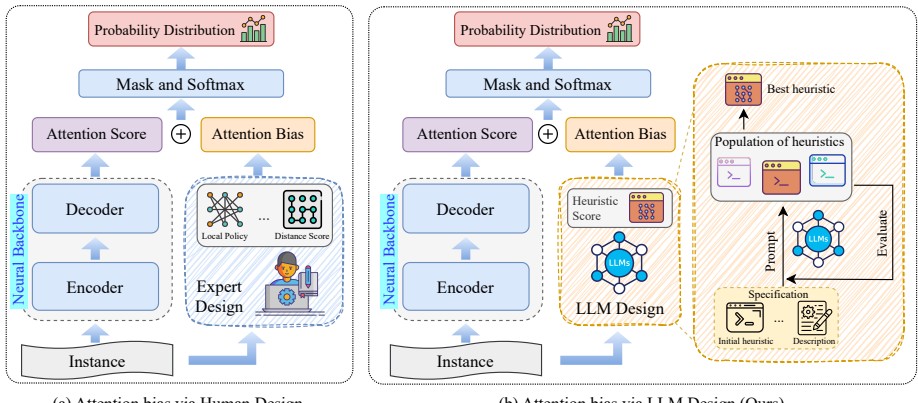

(a) Attention bias via Human Design    (b) Attention bias via LLM Design (Ours)

Figure 1: The diagrams illustrate the adaptation of neural solvers with attention bias designed by both human experts (left) and LLMs (right). The LLM-generated attention bias is then utilized to fine-tune pre-trained NCO models, enhancing their generalization capabilities for large-scale VRPs.

can be factorized into a chain of conditional probabilities as follows:

$$\mathbf{p}\left(\pi|\mathbf{s},\theta\right) = \prod_{t=1}^{l} \mathbf{p}\left(\pi_t|\pi_{1:t-1},\mathbf{s},\theta\right),\tag{1}$$

where $\mathbf{p}$ represents the policy parameterized by $\theta$, and $l$ represents the number of actions taken to complete the tour. In the case of TSP, $l = n$, while for CVRP, $l \geq n$ because the depot is visited at least once.

**Current Methods for Better Generalization.** Existing approaches to improving generalization in combinatorial optimization (CO) leverage divide-and-conquer strategies Fu et al. (2021b); Hou et al. (2023), problem symmetries Kim et al. (2022); Drakulic et al. (2023), and heavy decoder architectures Luo et al. (2023). Hybrid methods further enhance performance by integrating multiple strategies Zhou et al. (2023); Ye et al. (2024b); Wang et al. (2024); Gao et al. (2024). Among them, ELG achieves strong scalability by combining local topological heuristics with neural macro-level guidance. Despite their effectiveness, methods like ELG introduce significant complexity due to additional architectural and hyperparameter design, increasing computational costs and reliance on expert-driven heuristics.

## 3 METHODOLOGY

We propose a novel LLM-assisted fine-tuning strategy to enhance the generalization capabilities of attention-based NCO models on large-scale routing problems. Our method comprises two primary components: LLM-guided attention bias within neural models and fine-tuning the network using this adjusted attention score on a small number of instances with larger sizes than the training data. Specifically, the initial attention score of an NCO model is modified by adding an attention bias generated automatically by an LLM, as shown in Figure 1. This method can be integrated into any attention-based NCO model. In this study, we demonstrate its effectiveness in two representative encoder-decoder architectures, i.e., POMO Kwon et al. (2020) and LEHD Luo et al. (2023).

### 3.1 ATTENTION BIAS FOR CONSTRUCTIVE NEURAL SOLVER

For a problem instance $\mathbf{s}$ with $n$ node features $(\mathbf{s}_1, \ldots, \mathbf{s}_n)$ (e.g., the coordinates of $n$ nodes for TSP and additional demands for CVRP), a constructive model parameterized by $\theta$ generates the solution in an autoregressive manner. This model architecture is composed of a Transformer-based encoder and decoder. Whereas the encoder produces embeddings of all input nodes and the decoder produces the solution $\pi$ of $n$ nodes sequentially, one node at a time.

**Encoder.** The encoder without positional encoding transforms each node feature $\mathbf{s}_i$ to its initial embedding $\mathbf{h}_i^{(0)}$ through a learned linear projection. Then the initial embeddings $\{\mathbf{h}_1^{(0)}, \ldots, \mathbf{h}_n^{(0)}\}$ are fed into $L$ attention layers to get the final node embedding matrix $H^{(L)} = (\mathbf{h}_1^{(L)}, \ldots, \mathbf{h}_n^{(L)})$ with the embedding process $H^{(l)} = \text{AttentionLayer}(H^{(l-1)})$. As the encoder of our model is the same as the previous pre-trained neural methods, we refer the reader to the Appendix for details of the Attention layer.

**Decoder.** In this work, we define the ***attention score*** of a constructive NCO model as a vector $\mathbf{u}$ to calculate the node selection probability. For a light decoder model such as POMO, the decoder computes attention score as compatibility of the query and keys with all nodes, i.e.,

$$\mathbf{u} = \frac{\boldsymbol{q}^{\mathrm{T}}\mathbf{K}}{\sqrt{d}}, \boldsymbol{q} \in \mathbb{R}^{d \times 1}, \mathbf{K} \in \mathbb{R}^{d \times n},$$

where $\boldsymbol{q}$ represents the query vector, $\mathbf{K}$ represents the matrix of keys, and $d$ is the embedding dimension. The query $\boldsymbol{q}$ comes from the context embedding, and the keys $\mathbf{K}$ come from node embeddings. The final attention score is clipped using the $\tanh$ function with control parameter $C$. For the heavy decoder model such as LEHD, the attention score is computed by

$$\mathbf{u} = W_O \widetilde{\mathbf{H}}^{(L)},$$

where $W_O$ is a learnable matrix and $\widetilde{\mathbf{H}}^{(L)}$ is final embedding matrix as output of decoder.

We further introduce a generic concept of ***attention bias***, which can be represented by vector $\hat{\mathbf{u}}$. The attention bias can be aggregated with the model's attention score to adjust the probability selection of nodes in constructing the solution. The decoder sequentially constructs the solution in $n$ steps by selecting node by node. At each decoding step $t$, the most suitable node is selected and added to the current solution based on the selection probability $\mathbf{p}^t$, which is computed by using $\text{mask}$ and $\text{softmax}$ functions, i.e.,

$$\mathbf{p}^t = \mathbf{p}(\pi_t | \pi_{1:t-1}, \theta) = \text{softmax}(\text{mask}(\mathbf{u} + \hat{\mathbf{u}})) \tag{2}$$

where $\text{mask}$ function set the scores of invalid node selection as $-\infty$.

The addition of attention bias to the current attention score of a neural model is a straightforward method for refining the model's flexibility and fine-tuning it to achieve improved generalization without modifying the model architecture. We noticed that the attention bias $\hat{\mathbf{u}}$ can be designed and computed using distance score matrix Wang et al. (2024). However, this approach relies heavily on expert design, making it inflexible and potentially suboptimal. Unlike previous work, we leverage LLMs to automatically generate attention bias based on the topological features of VRP instances. This LLM-guided attention is then integrated with the original model's attention score to fine-tune existing neural models without the need for retraining from scratch, as detailed in the following subsections.

## 3.2 ATTENTION BIAS VIA LLM DESIGN

Attention bias, as previously mentioned, is an additional element that can be combined with attention score to facilitate and further customize the model fine-tuning process. In order to effectively design attention bias using LLMs, we harness the capabilities of evolutionary frameworks with LLMs, particularly EoH Liu et al. (2024), to generate and refine attention bias automatically. We initially examine a heuristic function $h(\boldsymbol{M}, \mathbf{s})$ that takes the distance matrix $\boldsymbol{M} \in \mathbb{R}^{n \times n}$ of $n$ nodes of a problem instance $\mathbf{s}$ as input and outputs a matrix $\mathbf{U}$ representing the attention bias of $n$ nodes, formally $\mathbf{U} = h(\boldsymbol{M}, \mathbf{s}), \mathbf{U} \in \mathbb{R}^{n \times n}$. Then, we utilize the evolution process to search this function under the following steps:

**Heuristic Representation.** Each heuristic function $h$ consists of three components: 1) a natural language description, 2) a code block in a predefined format, and 3) a fitness value. The function description in natural language provides a high-level overview of the heuristic. The fitness value $f(h)$ evaluates the heuristic's performance on a set of problem instances. An example of a heuristic to perform attention bias for TSP of 200 nodes (Pytorch code) is given in Figure 3 in the Appendix.

**Fitness Evaluation.** We evaluate the fitness of these heuristics on instances with larger sizes than those utilized for training the neural network model. For instance, if the neural model is pre-trained

with instances of 100 nodes, the evaluation dataset will comprise instances with either 150, 200, or 500 nodes. Specifically, we compute the aggregated attention score by combining the original model's attention score with the attention bias generated by the new heuristic. The updated attention score is then used to compute the node selection probability as in Eq. 2. The fitness value of the heuristic function is set as the average objective value of the constructed solutions for problem instances.

**Population Initialization.** We initialize a population $P$ of $N$ heuristics $h_1, \ldots, h_N$ by prompting LLMs using *Initialization prompt* to eliminate the need for expert knowledge. We generate $N$ initial heuristics by repeating the process $N$ times. The specifics of the initialization prompt for each TSP and CVRP problem are described in the corresponding subsections in the Appendix.

**Evolution Process.** We perform the evolutionary process to find the best heuristic by repeating the following $G$ iterations:

- Select parent heuristics from the current population to create offspring heuristics.
- Request LLM to produce a novel heuristic along with its related code implementation. The offspring heuristics are generated by using five evolution prompt strategies as exploration and modification operators. The details of these prompt strategies are described in the Appendix.
- The new heuristics are then evaluated on a set of evaluation instances to determine their fitness value.
- Add the new heuristic to the current population if both the heuristic and its code are valid.
- Select the top $N$ heuristics from the current population to form the new population for the next generation.

Once the LLM has designed the heuristic function for calculating the attention bias, we fine-tune the pre-trained neural model on larger-size instances to improve its scalability. Further details are provided in the following section.

## 3.3   MODEL FINE-TUNING

The attention bias designed by LLM can be immediately combined with pre-trained models to solve VRP problems. However, models trained on small, fixed-size datasets often suffer from limited generalization performance. Therefore, updating the weights of pre-trained models with larger-size datasets, guided by attention bias, is crucial for enhancing the stability and robustness of the model's generalization capability. We propose a new fine-tuning strategy based on LLM-guided attention bias to address this. Specifically, for a model pre-trained on 100-node instances, we fine-tune the model on varying-size instances within the range of 100 to 200 nodes jointly and normalize the loss function according to the number of nodes. This fine-tuning requires only a small amount of data, significantly reducing the resources and computational time compared to retraining the model from scratch, which is particularly costly, especially for RL-based models on large datasets.

After generating attention bias from LLM, we update vector $\mathbf{p}$ based on Equation equation 2 in each feed-forward. We modify the loss function of the neural models and then fine-tune the model by training on a small amount of data. Specifically, for POMO-LLM, the loss function is defined as the expectation of the cost $L(\pi)$ (total tour length) normalized by the number of nodes, i.e.,

$$\mathcal{L}(\theta|\mathbf{s}) = \mathbb{E}_{\mathbf{p}(\pi|\mathbf{s},\theta)}\big[\frac{L(\pi)}{n}\big] = \mathbb{E}_{\mathbf{p}(\pi|\mathbf{s},\theta)}[-R(\pi)], \tag{3}$$

where $R(\pi) = -\frac{L(\pi)}{n}$ is the total reward of solution $\pi$. Following POMO Kwon et al. (2020), we use multiple rollouts from different start nodes to get multiple trajectories, i.e., $\{\pi^1, \ldots, \pi^N\}$, in a single feed-forward and utilize the REINFORCE Williams (1992) algorithm with shared baseline to estimate the gradient of the loss. Specifically, the average reward of multiple rollouts, i.e., $b(\mathbf{s}) = \frac{1}{N}\sum_{j=1}^{N} R(\pi^j)$, is used as the shared baseline, and the gradient is estimated by:

$$\nabla_\theta \mathcal{L}(\theta|\mathbf{s}) \approx -\frac{1}{N}\sum_{i=1}^{N} \big(R\big(\pi^i\big) - b(\mathbf{s})\big) \nabla_\theta \log \mathbf{p}\big(\pi^i|\mathbf{s},\theta\big), \tag{4}$$

---

**Algorithm 1** Fine-tuning NCO model

---

**Input**: Pre-trained policy network parameter $\theta$ on training set of 100 nodes, number of epochs $E$, steps per epoch $T$, batch size $B$, training set $\{S_n\}, (n \in \mathbb{N}, n \in [100, 200])$
**Output**: Updated policy network $\theta$

1: **for** $epoch = 1, \ldots, E$ **do**
2:     $n \leftarrow \text{Random\_Integer}([100, 200])$
     // Training model on $S_n$ with mini-batch
3:     **for** $step = 1, \ldots, T$ **do**
4:        $\nabla_\theta \mathcal{L}(\theta) \leftarrow \frac{1}{B} \sum_{i=1}^{B} \nabla_\theta \mathcal{L}(\theta | \mathbf{s}_i \in S_n)$
5:        $\theta \leftarrow \text{Adam}(\theta, \nabla_\theta \mathcal{L}(\theta))$
6:     **end for**
7: **end for**
8: **return** updated policy $\theta$

---

where:

$$\mathbf{p}\left(\pi^i | \mathbf{s}, \theta\right) = \prod_{t=2}^{n} \mathbf{p}^t(\pi_t^i | \pi_{1:t-1}^i, \theta), \forall i \in [1, N]$$

Accordingly, fine-tuning an NCO model, e.g., POMO-LLM, is presented in Algorithm 1.

To fine-tune other models like LEHD-LLM, we simply need to adjust the loss function by regularizing it based on the number of nodes. The remaining optimization steps in fine-tuning model are similar to those used for POMO-LLM.

## 4 EXPERIMENTS AND RESULTS

In this section, we empirically verify the superiority of our proposed models against various NCO and classical solvers on two typical VRPs, i.e., TSP and CVRP, ranging from hundreds to several thousands of nodes. We outline the experimental settings and present results to assess our model's performance and validate significant components.

### 4.1 EXPERIMENTAL SETTINGS

#### 4.1.1 DATASETS.

To generate TSP and CVRP datasets, we follow the same data generation procedure and setting in previous work Kool et al. (2019). For fine-tuning our models, we use 10,000 instances for TSP and 5000 instances for CVRP, which are considerably smaller compared to the training data used in previous models such as POMO and LEHD. Since LEHD is trained using supervised learning, we use the Concorde Applegate et al. (2006b) solver to obtain optimal solutions for TSP and HGS Vidal (2022a) for CVRP labels. For testing, we utilize the test datasets available in previous work Luo et al. (2023) and well-known instances from TSPLib Reinelt (1991) and CVRPLib Uchoa et al. (2017) for TSP and CVRP, respectively.

#### 4.1.2 BASELINES

For classical solvers, we employ Concorde Applegate et al. (2006a), OR-Tools, and two non-learning heuristics, LKH3 Helsgaun (2017a) and HGS Vidal (2022a), as state-of-the-art (SOTA) non-neural methods for TSP and CVRP. For neural solvers, we compare our approach against POMO Kwon et al. (2020), LEHD Luo et al. (2023), MDAM Xin et al. (2021), Att-GCN+MCTS Fu et al. (2021a), BQ Drakulic et al. (2023), GLOP Ye et al. (2024b), and ELG Gao et al. (2024). Additionally, we evaluate our model against ReEvo Ye et al. (2024a), an automatic heuristic generation strategy using LLMs that are applied for attention reshaping Wang et al. (2024).

### 4.1.3 METRICS AND INFERENCE

We evaluate all methods based on optimality gap and inference time. The optimality gap measures the deviation from optimal solutions obtained via Concorde for TSP and LKH3 for CVRP. Since classical solvers run on a single CPU, their inference times are not directly comparable to learning-based methods executed on a GPU. For MDAM, POMO, LEHD, and ELG, we ran their official implementations with default settings on our test set. Results for Att-GCN+MCTS are cited from the original study, while BQ was reproduced and trained following its original paper. For POMO-LLM, we report results with and without ×8 instance augmentation (augx8 and no-aug). For LEHD-LLM, we provide results for both greedy inference and Random Re-Construct (RRC) with 50 iterations.

Table 1: Experimental results on TSP and CVRP on the generated instances. The asterisk (*) denotes the results of methods directly obtained from the original paper. The best results obtained by neural solvers are in **bold**.

| Method | TSP200 | Time | TSP500 | Time | TSP1000 | Time | CVRP200 | Time | CVRP500 | Time | CVRP1000 | Time |
|---|---|---|---|---|---|---|---|---|---|---|---|---|
| LKH/LKH3 | 0.000% | 4m | 0.000% | 22m | 0.000% | 2.4h | 0.000% | 22m | 0.000% | 1.8h | 0.000% | 2.1h |
| Concorde/HGS | 0.000% | 2m | 0.000% | 22m | 0.000% | 2.3h | -1.126% | 17m | -1.794% | 1.3h | -2.162% | 1.8h |
| OR-Tools | 3.618% | 10m | 4.682% | 38m | 4.885% | 2.9h | 6.894% | 15m | 9.112% | 38m | 11.662% | 50m |
| Att-GCN+MCTS* | 0.884% | 2m | 2.536% | 6m | 3.223% | 13m | - | - | - | - | - | - |
| MDAM bs50 | 1.996% | 2m | 10.065% | 3m | 20.375% | 20m | 4.304% | 1m | 10.498% | 4m | 27.814% | 18m |
| POMO no-aug | 2.394% | <1s | 25.090% | 2s | 43.623% | 6s | 6.097% | <1s | 30.164% | 2.5s | 144.664% | 4s |
| POMO augx8 | 1.622% | 2s | 23.093% | 10s | 41.810% | 0.6m | 5.022% | 2.5s | 20.377% | 10s | 128.885% | 0.8m |
| BQ greedy | 0.895% | 2s | 1.834% | 8s | 3.965% | 0.5m | 3.527% | 2s | 5.121% | 8s | 9.812% | 20s |
| BQ bs16 | 0.224% | 1m | 0.896% | 3m | 2.605% | 12.8m | 1.141% | 0.6m | 2.991% | 2.8m | 7.784% | 13.7m |
| LEHD greedy | 0.859% | 1.7s | 1.560% | 0.2m | 3.168% | 1.3m | 3.312% | 2.4s | 3.178% | 0.23m | 4.912% | 1.34m |
| LEHD RRC50 | 0.123% | 2.5m | 0.482% | 9.6m | 1.416% | 27m | 0.515% | 0.6m | 0.930% | 3.63m | 2.814% | 16.4m |
| ELG no-aug | 2.722% | 1.1s | 8.390% | 4.8s | 12.413% | 0.17m | 2.605% | 5.6s | 7.232% | 0.16m | 14.401% | 0.44m |
| ELG augx8 | 1.408% | 4.2s | 7.216% | 0.4m | 11.372% | 1.2m | 1.660% | 9s | 6.036% | 0.5m | 11.969% | 1.8m |
| POMO+ReEvo* | 4.02% | - | 24.32% | - | 29.08% | - | 10.48% | - | 25.7% | - | 218.22% | - |
| LEHD+ReEvo* | 0.74% | - | 1.55% | - | 2.97% | - | 3.30% | - | 2.94% | - | 4.76% | - |
| POMO-LLM no-aug | 1.364% | <1s | 4.219% | 3s | 9.794% | 7s | 2.379% | 1s | 3.132% | 2.7s | 11.599% | 4.2s |
| + augx8 | 0.888% | 2.4s | 3.479% | 0.3m | 9.003% | 0.8m | 1.665% | 2.6s | 2.599% | 10s | 9.314% | 0.8m |
| LEHD-LLM greedy | 0.766% | 1.8s | 1.375% | 0.2m | 2.427% | 1.3m | 2.727% | 2.4s | 1.946% | 0.24m | 3.456% | 1.33m |
| + RRC50 | **0.098%** | 2.7m | **0.368%** | 9.8m | **1.023%** | 28m | **0.307%** | 0.6m | **0.194%** | 3.65m | **1.399%** | 16.4m |

### 4.2 MAIN RESULTS

The main experimental results on uniformly distributed TSP and CVRP instances are summarized in Table 1. Compared to previous baselines like ELG, LEHD, POMO, and MDAM, our models, POMO-LLM and LEHD-LLM, consistently reduce the optimality gaps across all instance sizes. Notably, with just 50 RRC iterations, LEHD-LLM significantly outperforms other learning-based methods and OR-Tools within reasonable time, achieving near-optimal performance for TSP200, a gap of under 0.37% for TSP500, and around 1% for TSP1000. POMO-LLM also surpasses methods such as ELG, POMO, and MDAM, both with and without instance augmentation. Additionally, the greedy versions of LEHD-LLM and no-aug POMO-LLM deliver promising performance with fast inference times across all instances.

For CVRP, both LEHD-LLM and POMO-LLM demonstrate strong greedy inference performance across all instances. With just 50 RRC iterations, LEHD-LLM outperforms all learning-based methods on all CVRP instances. POMO-LLM also shows improved performance over ELG across most CVRP instances, except for the ELG aug method on CVRP200, where results are comparable. Additionally, POMO-LLM surpasses OR-Tools across all instances with significantly faster inference times while POMO struggles with poor generalization on large-scale instances such as CVRP500 and CVRP1000.

Overall, the experimental results for both TSP and CVRP demonstrate that LEHD-LLM and POMO-LLM enhance generalization and significantly reduce the performance gaps seen in previous models. Furthermore, their strong zero-shot generalization on large-scale instances highlights the effectiveness of LLMs in fine-tuning neural networks for improved scalability.

### 4.3 RESULTS ON TSPLIB AND CVRPLIB

In this section, we provide a detailed analysis of the generalization performance of various methods on TSPLib and CVRPLib benchmarks. The results for TSPLib, which includes instances ranging

from 100 to 5000 nodes, are presented in Table 2. For CVRPLib, we evaluate two datasets: Set-X, with instances from 100 to 1000 nodes, and Set-XXL, with instances from 3000 to 7000 nodes. The corresponding results are shown in Tables 3 and 5, respectively.

Table 2: Experimental results on TSPLib.

| Method | [100-200] | [200-500] | [500-1k] | [1k-5k] |
|---|---|---|---|---|
| POMO no-aug | 3.74% | 11.46% | 22.57% | 40.32% |
| POMO augx8 | 2.49% | 9.57% | 20.42% | 36.85% |
| BQ greedy | 2.68% | 3.18% | 8.31% | 42.57% |
| BQ bs16 | 1.32% | 2.18% | 5.52% | 36.73% |
| LEHD greedy | 2.37% | 2.64% | 5.23% | 11.55% |
| LEHD RRC50 | **0.48%** | **0.79%** | **2.17%** | **6.48%** |
| ELG no-aug | 2.54% | 6.93% | 9.24% | 12.74% |
| ELG augx8 | 1.30% | 3.88% | 8.73% | 11.33% |
| POMO-LLM no-aug | 3.45% | 6.33% | 9.17% | 13.12% |
| POMO-LLM augx8 | 2.12% | 4.19% | 8.02% | 12.18% |
| LEHD-LLM greedy | 2.33% | 3.08% | 3.41% | 9.53% |
| LEHD-LLM RRC50 | **0.32%** | **0.63%** | **1.83%** | **5.56%** |

Table 3: Experimental results on CVRPLib on Set-X dataset.

| Method | [100-200] | [200-500] | [500-1k] |
|---|---|---|---|
| POMO no-aug | 9.743% | 19.164% | 57.227% |
| POMO augx8 | 6.889% | 15.029% | 40.904% |
| LEHD greedy | 11.345% | 9.452% | 17.737% |
| LEHD RRC50 | 4.908% | 4.732% | 8.273% |
| ELG no-aug | 6.253% | 7.576% | 10.024% |
| ELG augx8 | **4.510%** | 5.524% | 7.805% |
| POMO-LLM no-aug | 8.600% | 10.604% | 23.154% |
| POMO-LLM augx8 | 5.948% | 8.507% | 15.256% |
| LEHD-LLM greedy | 10.922% | 9.393% | 16.439% |
| LEHD-LLM RRC50 | **4.839%** | **4.624%** | **7.678%** |

Table 2 shows the test results on real-world TSPLib instances with different sizes and distributions. We report the optimal gap values for LEHD-LLM and POMO-LLM on both their respective versions. It can be seen that LEHD-LLM with RRC 50 iterations still achieves the best results on all instances with different sizes. The results are significantly improved over ELG, BQ, LEHD, and POMO. POMO-LLM still significantly improves the performance of POMO, especially on large data sets. The gap reduction is around 12% for instances with 500-1000 nodes and a remarkable 24-27% for instances with 1000 to several thousand nodes.

The experimental results for CVRPLib on Set-X, as shown in Table 3, indicate that LEHD-LLM outperforms other methods across most instance sizes, except for the 100-200 nodes, where it achieves competitive results with ELG. The additional results on CVRPLib Set-XXL are shown in Table 5. Since POMO-based methods exhibit significant performance degradation when the number of nodes exceeds 1000, we do not report their results on the XXL dataset. As shown in Table 5, LEHD-LLM

Table 4: Experimental results on CVRPLib on Set-XXL dataset.

| Method | Leuven1 | Leuven2 | Antwerp1 | Antwerp2 |
|---|---|---|---|---|
| LEHD greedy | 16.60% | 34.85% | 14.66% | 22.77% |
| LEHD RRC50 | 11.98% | 28.46% | 10.78% | 18.16% |
| GLOP | 16.90% | 21.80% | 20.30% | 19.40% |
| GLOP-LKH3 | 16.60% | **21.10%** | 19.30% | 19.40% |
| ELG no-aug | 16.45% | 23.26% | 13.65% | 26.13% |
| ELG augx8 | 12.26% | 21.60% | 12.26% | 17.74% |
| LEHD-LLM greedy | 15.12% | 30.35% | 13.61% | 17.77% |
| LEHD-LLM RCC50 | **11.68%** | 25.97% | **9.64%** | **14.85%** |

RRC50 delivers the best performance on 3 out of the 4 datasets, with the remaining top result achieved by GLOP. Compared to other methods, including ELG and GLOP, LEHD-LLM substantially narrows the optimality gap for large datasets such as Leuven1, Antwerp1, and Antwerp2. The obtained results demonstrate the promising generalization ability of our proposed model when handling large-scale instances.

## 4.4 ABLATION STUDIES

The effectiveness of the proposed models lies in the combination of attention bias generated by the LLM and the fine-tuning process. To investigate the model's efficiency further, we conduct detailed ablation studies as follows.

### 4.4.1 EFFECTS OF ATTENTION BIAS

We replace the attention bias generated by the LLM with direct information from the normalized distance matrix following Wang et al. (2024). Then, we continue fine-tuning the model with the attention bias on the POMO model and report the results as shown in Figure 2(a). The results show that fine-tuning POMO-LLM with attention bias is more effective than one with a normalized distance matrix.

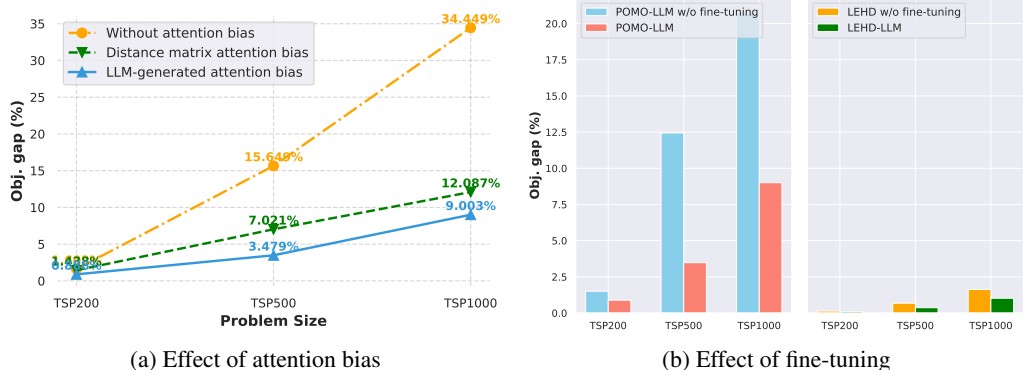

(a) Effect of attention bias
(b) Effect of fine-tuning

Figure 2: (a) Performance of fine-tuning POMO-LLM model with different attention biases and without any on the TSP test instances. (b) Performance of POMO-LLM and LEHD-LLM models with and without (w/o) fine-tuning strategy.

### 4.4.2 EFFECTS OF FINE-TUNING STRATEGY

We show the results of NCO models with and without the proposed fine-tuning strategy on TSP test instances in Figure 2(b). It confirms that the performance of solely using the LLM-generated attention biases without fine-tuning is not as good as using fine-tuning, especially in POMO-LLM.

### 4.4.3 EFFECTS OF DIVERSE SCALES FOR FINE-TUNING

To evaluate the impact of fine-tuning the model on different data sizes, we compared the performance of POMO-LLM fine-tuning on TSP instances with fixed size (200 nodes) and POMO-LLM with fine-tuning on varying sizes randomly selected from $[100 - 200]$ nodes. The results are presented in Table 5.

The experimental results show that fine-tuning the TSP model on instances of varying sizes enhanced performance compared to using fixed sizes.

Table 5: Effect of problem size on fine-tuning POMO-LLM for TSP.

| Method | TSP200 | TSP500 | TSP1000 |
|---|---|---|---|
| POMO-LLM fixed size | 0.972% | 4.216% | 11.782% |
| POMO-LLM varying size | **0.888%** | **3.479%** | **9.003%** |

### 4.4.4 ATTENTION BIAS DESIGN USING DIFFERENT LLMS

We evaluate the effectiveness of attention bias generation using four widely used LLMs: GPT-4o mini, Llama, Mixtral, and Gemma. All experiments are conducted under identical settings on the TSP problem to ensure a fair comparison. Our experimental results indicate that integrating LLM-generated attention bias consistently improves the performance of NCO models. Among the evaluated models, GPT-4o mini achieves the best performance, demonstrating superior generalization and solution quality compared to the other LLMs.

### 4.4.5 COMPLEXITY ANALYSIS

We reiterate that the complexity of the NCO models enhanced by LLM-generated attention bias is negligibly greater than the original NCO models. In addition, the complexity of generating attention bias based on the EoH framework mainly lies in the evolutionary search process. In each generation, we limit the time for each request to the pre-trained LLM to 10 seconds through experimentation to achieve the most appropriate results and runtime.

## 5 CONCLUSION

In conclusion, this work demonstrates that integrating Large Language Models (LLMs) with Neural Combinatorial Optimization (NCO) methods enhances the performance of neural solvers for large-

scale Vehicle Routing Problems (VRPs). By integrating LLM-guided attention bias with neural models and fine-tuning them on limited data, we improve both flexibility and scalability without requiring extensive retraining. The experimental results validate that LLM-assisted neural solvers significantly enhance the state-of-the-art for solving the Travelling Salesman Problem (TSP) and the Capacitated Vehicle Routing Problem (CVRP), demonstrating superior performance on both synthetic and real-world problems with thousands of nodes. In the future, we could explore diverse LLM architectures, adapt to VRPs with complex constraints and other combinatorial optimization problems to improve scalability and efficiency, enhance interpretability for decision support, and benchmark against traditional heuristics to expand the applicability and effectiveness of LLM-powered neural solvers.

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

## A DETAILS OF OUR METHOD

### A.1 CONSTRUCTIVE NEURAL SOLVERS

#### A.1.1 ENCODER.

Given a problem instance $\mathbf{s}$ with $n$ node features $(\mathbf{s}_1, \ldots, \mathbf{s}_n)$, the encoder transforms each node feature $\mathbf{s}_i$ to its initial embedding $\mathbf{h}_i^{(0)}$ through a learned linear projection with parameters $W^{\mathbf{s}}$ and $\mathbf{b}^{\mathbf{s}}$: $\mathbf{h}_i^{(0)} = W^{\mathbf{s}} \mathbf{s}_i + \mathbf{b}^{\mathbf{s}}$. Then the initial embeddings $\{\mathbf{h}_1^{(0)}, \ldots, \mathbf{h}_n^{(0)}\}$ are fed into $L$ attention layers to get the final node embedding matrix $H^{(L)} = (\mathbf{h}_1^{(L)}, \ldots, \mathbf{h}_n^{(L)})$ with the embedding process $H^{(l)} = \text{AttentionLayer}(H^{(l-1)}), l \in \{1, \ldots, L\}$.

#### A.1.2 ATTENTION LAYER.

The attention layer comprises two sub-layers: the multi-head attention (MHA) sub-layer and the feed-forward (FF) sub-layer Kool et al. (2019); Kwon et al. (2020); Luo et al. (2023). In addition, each sublayers add skip-connection and normalization, i.e., batch normalization (BN) for POMO Kwon et al. (2020) or layer normalization (LN). Let $H^{(l-1)} = (\mathbf{h}_1^{(l-1)}, \ldots, \mathbf{h}_n^{(l-1)})$ be the input of the $l$-th attention layer for $l = 1, \ldots, L$, the output of the attention layer in terms of the $i$-th node is calculated as:

$$
\begin{aligned}
\hat{\mathbf{h}}_i^{(l)} &= \text{BN}^l \left( \mathbf{h}_i^{(l-1)} + \text{MHA} \left( \mathbf{h}_i^{(l-1)}, H^{(l-1)} \right) \right), \\
\mathbf{h}_i^{(l)} &= \text{BN}^l \left( \hat{\mathbf{h}}_i^{(l)} + \text{FF} \left( \hat{\mathbf{h}}_i^{(l)} \right) \right).
\end{aligned}
\tag{5}
$$

We refer the readers to AM Kool et al. (2019) for more details of MHA and FF sublayers.

Note that, for the LEHD Luo et al. (2023), the normalization is removed from our model to enhance generalization performance, and the light encoder has only one attention layer while the heavy decoder has $L$ attention layers.



**Function description:** The new algorithm computes the heuristics matrix by calculating a normalized version of the distances, where closer distances have higher positive values and farther distances yield negative values, thus assigning promising edges higher values and undesirable edges lower values.

**Code:**
```python
def heuristic(distance_matrix):
    dist_tensor = torch.tensor(distance_matrix, dtype=torch.float32)
    # Calculate the maximum distance for each node (row-wise maximum)
    max_distances = dist_tensor.max(dim=1)[0]
    # Calculate contribution scores for each edge
    heuristics_matrix = max_distances.unsqueeze(1) - dist_tensor
    return heuristics_matrix
```
**Fitness value:** 10.74865



Figure 3: An example of a heuristic representation.

## A.2   ATTENTION BIAS VIA LLM DESIGN

**Initialization prompts.**

We use LLMs to generate the initial $N$ heuristics as follows: First, we inform the LLMs about the heuristic design for TSP and CVRP, then instruct them to create new heuristics based on this description and implement them in code. Specifically, we direct the LLMs to generate PyTorch code for seamless integration with the fine-tuning of NCO models. Following EoH Liu et al. (2024), we design the prompt engineering for heuristics initialization consists of four mains components as following:

- **Task Description**: This informs LLMs about the problem.
- **Strategy-Specific Prompt**: These prompts guide the LLMs to reason over the information and generate new heuristics, along with their code implementation.
- **Expected Output**: Specifies that the LLM should produce both a heuristic description and its code implementation, typically in Pytorch, with clearly defined names, inputs, and outputs for easy identification within the Evolution of Heuristics (EoH) framework.
- **Note**: Provides extra instructions to enhance the LLM's efficiency and robustness. This could include suggesting specific input/output types and discouraging overly lengthy explanations.

Examples of Initialization Prompts for attention bias design on TSP and CVRP are illustrated in Figure 4.

**Evolution prompts to create new heuristics.**

Following EoH Liu et al. (2024), we use five strategies for generating new heuristics, categorized into two groups: Exploration (E1, E2) and Modification (M1, M2, M3).

Exploration Strategies:

- E1: Create heuristics that differ as much as possible from selected parents to explore new ideas.
- E2: Develop heuristics that share core ideas with selected parents but introduce new elements.

Modification Strategies:

- M1: Improve a selected heuristic for better performance.
- M2: Adjust the parameters of a chosen heuristic to create a new variant.
- M3: Simplify a heuristic by removing redundant components after analyzing its structure.

The prompt engineering for evolution to create new heuristics includes the same four components as in initialization, with an additional component called parent heuristics:

---

**Initialization Prompt for TSP**

I need assistance in designing a new heuristic to improve the solution of the Traveling Salesman Problem (TSP) by incorporating insights from prior heuristics. The TSP requires finding the shortest path that visits all given nodes and returns to the starting node. The new algorithm computes a heuristic matrix where the distances between nodes are normalized. In this matrix, closer distances are assigned higher positive values, while farther distances are given negative values. This approach aims to prioritize promising edges by assigning them higher scores and to de-prioritize less desirable edges by giving them lower scores.

Please design a new heuristic.

**First,** describe your new heuristic and main steps in one sentence. The description must be inside a brace.

**Next,** implement it in Pytorch as a function named 'heuristic'. This function should accept one input: 'distance matrix'. The function should return one output: 'heuristic matrix'.

Note that 'distance matrix' is a tensor, and 'heuristic matrix' is a tensor with the same shape as 'distance matrix.' All are Torch tensors. Do not give additional explanations.

**Initialization Prompt for CVRP**

I need assistance in designing a new heuristic to improve the solution of the Capacitated Vehicle Routing Problem (CVRP) by incorporating insights from prior heuristics. The CVRP requires finding the shortest path that visits all given nodes and returns to the starting node, with the additional constraint that each node has a demand and each vehicle has a capacity. The total demand of the nodes visited by a vehicle cannot exceed its capacity. If the total demand exceeds the vehicle's capacity, the vehicle must return to the starting node before continuing.

Please design a new heuristic.

**First,** describe your new heuristic and main steps in one sentence. The description must be inside a brace.

**Next,** implement it in Pytorch as a function named 'heuristic'. This function should accept two inputs: 'distance matrix' and 'node_demands'. The function should return one output: 'heuristic matrix'.

Note that 'distance matrix' and 'node_demands' are tensors, and 'heuristic matrix' is a tensor with the same shape as 'distance matrix.' All are Torch tensors. Do not give additional explanations.

Figure 4: Two examples of initialization prompts used for TSP and CVRP to create attention bias.

- **Parent heuristics**: This component includes parent heuristics to enable in-context learning over both the linguistic description and the code implementation of the heuristic.

Illustrated examples of two evolution prompts for exploration operator E1 and modification operator M1 for TSP are shown in Figure 5.

## B    EXPERIMENTAL SETTINGS

### B.1    MODEL SETUP AND TRAINING

Our models leverage the same network architecture as previous representative attention-based NCO architectures, i.e., POMO Kwon et al. (2020) and LEHD Luo et al. (2023). The training process is divided into two stages: pre-training and fine-tuning. For fair comparisons, the total number of training steps for the proposed models is the same as those of the baseline models. We pre-trained the POMO and LEHD models on TSP100 and CVRP100 instances and then fine-tuned these models on a small dataset of larger-scale instances, e.g., 150 and 200 nodes, with additional attention bias generated by LLM. We use the pre-trained LLM GPT-4o mini to generate attention bias.

### B.2    HYPERPARAMETERS

For designing attention bias using LLMs through the evolution process, we set the number of generations (iterations) $G$ as 20 and the population size $N$ as 10. We use pre-trained NCO models for TSP and CVRP with 100 nodes combined with attention bias to evaluate the fitness of heuristics on larger-size instances to enable the large-scale generation capability of NCO models. Specifically, we use 128 instances of 200 nodes for both TSP and CVRP to evaluate the heuristics designed by LLMs. The evaluation is performed in parallel using GPU for a faster evolution process.

For fine-tuning, we used pre-trained TSP and CVRP models on one million instances of size 100, with published checkpoints of POMO Kwon et al. (2020) and LEHD Luo et al. (2023). For TSP, we

Evolution Prompt for E1:

I need assistance in designing a new heuristic to improve the solution of the Traveling Salesman Problem (TSP) by incorporating insights from prior heuristics. The TSP requires finding the shortest path that visits all given nodes and returns to the starting node. The new algorithm computes a heuristic matrix where the distances between nodes are normalized. In this matrix, closer distances are assigned higher positive values, while farther distances are given negative values. This approach aims to prioritize promising edges by assigning them higher scores and to de-prioritize less desirable edges by giving them lower scores.

I have five existing algorithms with their codes as follows:

No.1 Heuristic description:

and the corresponding code are:

...

No.5 Heuristic description:

and the corresponding code are:

Please help me create a new algorithm that has a totally different form from the given ones.

**First,** describe your new heuristic and main steps in one sentence. The description must be inside a brace.

**Next,** implement it in Pytorch as a function named 'heuristic'. This function should accept one input: 'distance matrix'. The function should return one output: 'heuristic matrix'.

Note that 'distance matrix' is a tensor, and 'heuristic matrix' is a tensor with the same shape as 'distance matrix.' All are Torch tensors. Do not give additional explanations.

Evolution Prompt for M1:

I need assistance in designing a new heuristic to improve the solution of the Traveling Salesman Problem (TSP) by incorporating insights from prior heuristics. The TSP requires finding the shortest path that visits all given nodes and returns to the starting node. The new algorithm computes a heuristic matrix where the distances between nodes are normalized. In this matrix, closer distances are assigned higher positive values, while farther distances are given negative values. This approach aims to prioritize promising edges by assigning them higher scores and to de-prioritize less desirable edges by giving them lower scores.

I have one algorithm with its code as follows:

Algorithm description:

Code:

Please assist me in creating a new algorithm that has a different form but can be a modified version of the algorithm provided.

**First**, describe your new algorithm and main steps in one sentence. The description must be inside a brace.

**Next**, implement it in Pytorch as a function named 'heuristic'. This function should accept one input: 'distance matrix'. The function should return one output: 'heuristic matrix'.

Note that 'distance matrix' is a tensor, and 'heuristic matrix' is a tensor with the same shape as 'distance matrix.' All are Torch tensors. Do not give additional explanations.

Figure 5: Two examples of two evolution prompts for exploration operator E1 and modification operator M1 for TSP.

fine-tuned the models on 10,000 samples with random sizes ranging from 100 to 200 for POMO-LLM and on sizes 100, 150, and 200 for LEHD-LLM. For CVRP, the models were fine-tuned on 5,000 samples of fixed size 200 for both models.

We used the Adam Kingma & Ba (2015) optimizer with an initial learning rate of $10^{-4}$. Following POMO, we set the weight decay to $10^{-6}$ and trained the models for 200 epochs for both TSP and CVRP. The scaling parameter $C$ for clipping the attention score was set to $50$ for faster convergence according to Gao et al. (2024); Jin et al. (2023). In line with LEHD, we applied a learning rate decay of $0.97$ per epoch for the TSP model and $0.9$ per epoch for the CVRP model. Additionally, LEHD-LLM was trained for $150$ epochs for TSP and $40$ epochs for CVRP. Both POMO-LLM and LEHD-LLM are fine-tuned with a batch size of $512$.

## B.3 HARDWARE

In all experiments, we train and test our models on a single GPU, i.e., NIVIDA A100-SXM4 with 40GB memory. We also test other learning-based methods on the same machine. For non-learning methods, we run them on the CPU, i.e., AMD R7 5800H 64 cores.

