# OpenReview forum: "Large Language Models powered Neural Solvers for Generalized Vehicle Routing Problems"
_ICLR.cc/2025/Workshop/AgenticAI — ICLR 2025 Workshop AgenticAI Oral_

### Official Review · Reviewer_K7YX · 2025-03-03

**Rating:** 9
**Confidence:** 4

**Review:**

This paper introduces a novel LLM-guided attention bias mechanism for model fine-tuning, aiming to enhance the generalization capabilities of neural combinatorial optimization (NCO) models.

Extensive experimental results demonstrate that the proposed approach achieves state-of-the-art performance on the Traveling Salesman Problem (TSP) and the Capacitated Vehicle Routing Problem (CVRP) across various problem scales. Additionally, the method exhibits strong generalization capabilities, effectively solving real-world TSPLib and CVRPLib instances.

Areas for Improvement:
- It would be better if the paper could analyze how the number of iterations (G) and the population size (N) impact the quality of the attention bias.
- It would be better if the paper could provide qualitative examples of the generated heuristic and analyze why these LLM generated heuristics provide better attention scores compared to other expert-designed heuristics.
- The paper should use parenthetical citations when references are not used as nouns.

---

### Official Review · Reviewer_zz3F · 2025-03-04
**An effective fine-tuning method for vehicle routing problems**

**Rating:** 7
**Confidence:** 4

**Review:**

This paper addresses the challenge of scaling Neural Combinatorial Optimization (NCO) for large Vehicle Routing Problem (VRP). It proposes an LLM-driven fine-tuning approach that integrates automatically generated attention bias into pre-trained neural solvers. By leveraging LLMs to design heuristics and refining models on diverse instance sizes, the method enhances generalization without retraining from scratch. Experiments on TSP and CVRP show state-of-the-art performance of the proposed method.

Strengths:
1. This paper proposes a "plug-in" method which can be easily combined with existing machine learning models for VRP. Therefore, the method may provide further improved performance suppose there are better foundation models for VRP, hence the impact of this paper is high.

2. The proposed method is a fine-tuning method in nature. It requires less time to implement than training a model from scratch when the data changes. Besides, adding bias introduces negligible computational cost to the initial foundation model to which the method is applied. Therefore, the proposed method has a high time efficiency.

3. This paper conducts well-designed experiments to verify the effectiveness of its proposed method. Results show that the proposed method achieves superior performance.

Weaknesses:
1. This paper relies on EoH[1] method to generate the attention bias. The difference from naive application of EoH is that this paper uses LLMs to initialize the heuristics. It may be better if analysis is provided on the benefit of such initialization vs for example random initialization or other possible initializations.

[1] Liu, Fei, et al. "Evolution of Heuristics: Towards Efficient Automatic Algorithm Design Using Large Language Model." International Conference on Machine Learning. PMLR, 2024.

---

### Official Review · Reviewer_NyHE · 2025-03-05
**Good Paper**

**Rating:** 7
**Confidence:** 4

**Review:**

This paper presents a Large Language Model (LLM)-powered neural solver for generalized Vehicle Routing Problems (VRPs), integrating LLM-generated heuristics with Neural Combinatorial Optimization (NCO). Traditional constructive NCO methods struggle with large-scale VRPs due to their reliance on attention-based node selection, which does not scale well. The proposed method fine-tunes existing neural solvers by introducing LLM-guided attention bias, derived through an evolutionary process that extracts structural features from VRP instances. This attention bias enhances the model’s flexibility and generalization without modifying its architecture. Experimental results on Traveling Salesman Problem (TSP) and Capacitated Vehicle Routing Problem (CVRP) show that the approach outperforms state-of-the-art solvers (e.g., POMO, LEHD, ELG) in both synthetic and real-world datasets (TSPLIB, CVRPLIB), achieving superior scalability and efficiency for large-scale combinatorial optimization.

Suggested Improvements:
1. The paper does not provide an in-depth analysis of how LLM-generated heuristics influence decision-making.
2. The study mainly compares against other NCO-based solvers, but real-world logistics often rely on classical heuristics (e.g., LKH, HGS, OR-Tools).

---

### Decision · Program_Chairs · 2025-03-05

Accept (Oral)